# Metabolomics Analysis Reveals Molecular Signatures of Metabolic Complexity in Children with Hypercholesterolemia

**DOI:** 10.3390/nu15071726

**Published:** 2023-03-31

**Authors:** Pei-Shin Gu, Kuan-Wen Su, Kuo-Wei Yeh, Jing-Long Huang, Fu-Sung Lo, Chih-Yung Chiu

**Affiliations:** 1Division of Pediatric Endocrinology, Department of Pediatrics, Chang Gung Memorial Hospital at Linkou, Chang Gung University College of Medicine, Taoyuan 333, Taiwan; 2Graduate Institute of Clinical Medical Sciences, College of Medicine, Chang Gung University, Taoyuan 333, Taiwan; 3Department of Pediatrics, Chang Gung Memorial Hospital at Keelung, Chang Gung University College of Medicine, Taoyuan 333, Taiwan; 4Department of Pediatrics, Chang Gung Memorial Hospital at Linkou, Chang Gung University College of Medicine, Taoyuan 333, Taiwan; 5Department of Pediatrics, New Taipei Municipal TuCheng Hospital, Chang Gung Memorial Hospital, Chang Gung University College of Medicine, Taoyuan 333, Taiwan; 6Clinical Metabolomics Core Laboratory, Chang Gung Memorial Hospital at Linkou, Taoyuan 333, Taiwan

**Keywords:** childhood hypercholesterolemia, glutamic acid, tyrosine, metabolomics

## Abstract

Despite the importance of hypercholesterolemia in children, it is overlooked, and there are currently few metabolomics-based approaches available to understand its molecular mechanisms. Children from a birth cohort had their cholesterol levels measured with the aim of identifying the metabolites for the molecular biological pathways of childhood hypercholesterolemia. One hundred and twenty-five children were enrolled and stratified into three groups according to cholesterol levels (acceptable, <170 mg/dL, *n* = 42; borderline, 170–200 mg/dL, *n* = 52; and high, >200 mg/dL, *n* = 31). Plasma metabolomic profiles were obtained by using ^1^H-nuclear magnetic resonance (NMR) spectroscopy, and partial least squares-discriminant analysis (PLS-DA) was applied using the MetaboAnalyst 5.0 platform. Metabolites significantly associated with different cholesterol statuses were identified, and random forest classifier models were used to rank the importance of these metabolites. Their associations with serum lipid profile and functional metabolic pathways related to hypercholesterolemia were also assessed. Cholesterol level was significantly positively correlated with LDL-C and Apo-B level, as well as HDL-C and Apo-A1 level separately, whereas HDL-C was negatively correlated with triglyceride level (*p* < 0.01). Eight metabolites including tyrosine, glutamic acid, ornithine, lysine, alanine, creatinine, oxoglutaric acid, and creatine were significantly associated with the different statuses of cholesterol level. Among them, glutamic acid and tyrosine had the highest importance for different cholesterol statuses using random forest regression models. Carbohydrate and amino acid metabolisms were significantly associated with different cholesterol statuses, with glutamic acid being involved in all amino acid metabolic pathways (FDR-adjusted *p* < 0.01). Hypercholesterolemia is a significant health concern among children, with up to 25% having high cholesterol levels. Glutamic acid and tyrosine are crucial amino acids in lipid metabolism, with glutamic-acid-related amino acid metabolism playing a significant role in regulating cholesterol levels.

## 1. Introduction

Dyslipidemia is a medical condition characterized by abnormal levels of lipids in the blood, such as cholesterol and triglycerides. It is also known as one of the strongest risk factors for cardiovascular disease, with the increasing fact that atherosclerosis begins in childhood. In clinics, there is growing attention regarding recognizing and treating of dyslipidemia in childhood to prevent or delay cardiovascular events in adulthood [1,2,3]. The management of dyslipidemia in childhood typically involves lifestyle changes such as increased physical activity, a healthy diet, and weight management. However, early detection and intervention are crucial in managing dyslipidemia in childhood.

Total cholesterol, the sum of non–high-density lipoprotein cholesterol (non–HDL-C) and HDL cholesterol (HDL-C), is the most often discussed factor for pediatric dyslipidemia [4]. Other lipid markers, such as low-density lipoprotein (LDL) cholesterol, triglycerides, and apolipoprotein B (Apo-B), are also important in the diagnosis and management of dyslipidemia. Elevated levels of LDL cholesterol and triglycerides, as well as low levels of HDL cholesterol, are associated with an increased risk of cardiovascular disease [5]. Elevated levels of Apo-B are strongly associated with an increased risk of cardiovascular disease, and some experts argue that Apo-B may be a more accurate predictor of cardiovascular risk than LDL cholesterol [6]. Screening of lipid profiles and early intervention in childhood can help prevent or delay the onset of dyslipidemia and reduce the risk of cardiovascular disease and other complications later in life.

In clinical practice, attention has been given to a hereditary form of pediatric hypercholesterolemia known as familial hypercholesterolemia. This condition is commonly inherited in an autosomal dominant pattern and is believed to affect approximately 1 in 250 individuals globally, rendering it one of the most frequently occurring genetic disorders [7]. Familial hypercholesterolemia is the result of mutations in genes responsible for the production of receptors that facilitate the removal of cholesterol from the bloodstream, leading to excessively high levels of cholesterol in the blood from birth. A recent study has explored the molecular characteristics of this genetic pediatric hypercholesterolemia [8]. However, the molecular mechanism of hypercholesterolemia in healthy children from birth remains understudied.

Despite the few reports, several molecular pathways have been implicated in the development of hypercholesterolemia. One major key mechanism of hypercholesterolemia involves the overproduction of low-density lipoprotein particles by the liver, which can lead to an accumulation of cholesterol in the blood [9]. In addition, chronic inflammation can lead to changes in lipid metabolism, including alterations in the production of oxidized lipids and changes in the composition of lipoproteins [10]. There is also growing evidence suggesting an association between specific amino acids and hypercholesterolemia. For example, increased levels of branched-chain amino acids (BCAAs), linked to insulin resistance, have reported to be associated with an increased risk of hypercholesterolemia and cardiovascular disease [11]. Holistically, the molecular mechanisms underlying hypercholesterolemia are complex and involve multiple pathways and factors.

A new platform of metabolomics has emerged as a powerful tool for studying human health, allowing for the simultaneous measurement of thousands of small molecules in biological samples, such as blood, urine, and tissue [12]. This technology has created opportunities to understand the underlying molecular mechanisms of various human diseases. It has also been applied to a wide range of health conditions, including metabolic disorders, cardiovascular disease, cancer, and neurological disorders, providing novel insights into disease mechanisms and identifying new biomarkers for diagnosis and therapy [13].

Metabolomics is utilized to analyze the specific biochemical molecules and metabolic pathways of an organism. Nuclear magnetic resonance (NMR) spectroscopy is a powerful analytical technique used in metabolomics research, known for its high reproducibility and ability to provide high-throughput molecular identification. Blood contains a wide range of metabolically active compounds and is commonly used to estimate the body’s cholesterol status, making it suitable for investigating the systemic biological metabolism and lipids.

This study aimed to identify the metabolic signatures of different cholesterol statuses and their relationship with the lipid profile in children using ^1^H-NMR spectroscopy. A comprehensive investigation of using a metabolomics-based approach and further functional pathway analysis can not only gain insight into the molecular mechanism of lipid metabolism but also provide potential health strategies for childhood hypercholesterolemia.

## 2. Materials and Methods

### 2.1. Study Population

A prospective cross-sectional controlled study was conducted to investigate the metabolomic profiles of blood in children with cholesterol levels. Children recruited in a birth cohort launched in Taiwan and who had completed a 7-year follow-up period since birth were enrolled into this study. Blood samples were obtained at the age of 7 years old. Children with any chronic health conditions including congenital heart disease, congenital chromosome or genetic anomaly, and endocrine disorders such as diabetes or thyroid disease were excluded. In addition, to minimize the potential con-founding effects of genetic familial hypercholesterolemia, children with a family health history of early heart disease or hypercholesterolemia were excluded from the study. Based on the guidelines from the American College of Cardiology (ACC), these children were stratified into 3 groups according to cholesterol level (acceptable, <170 mg/dL; borderline, 170–200 mg/dL; and high, >200 mg/dL) [14]. Information regarding demographic data including the children’s age, sex, and body mass index (BMI) was collected.

### 2.2. Serum Cholesterol and Lipid Profile Measurement

Blood samples were collected at the outpatient clinics after the subjects had fasted for at least 8 hours. The collected samples were frozen immediately and stored at −80 °C until required. Total cholesterol concentrations as well as triglyceride, low-density lipoprotein (LDL), and high-density lipoprotein (HDL) levels were measured by an automated enzymatic colorimetric assay (Hitachi LST 008, Tokyo, Japan). Apolipoprotein A1 (Apo-A1) and Apo-B levels were measured using an immunoturbidimetric assay with an automatic biochemical analyzer. Serum adiponectin and leptin levels were measured by enzyme-linked immunosorbent assay (ELISA) (Tecan Sunrise, Mannedorf, Switzerland).

### 2.3. H–NMR Spectroscopy

As previously described [15], plasma samples were prepared prior to spectrum acquisition. The prepared 500 μL plasma was mixed with 500 μL phosphate buffer containing 0.08% 3-(trimethylsilyl)-propionic-2,2,3,3-d4 acid sodium salt (TSP) as an internal chemical shift reference. A standard 5mm NMR tube was later filled with an aliquot of 600 μL of the mixed preparation for analysis. ^1^H-NMR spectra were obtained from a Bruker Advance 600 MHz spectrometer (Bruker-Biospin GmbH, Karlsruhe, Germany) at the Chang Gung Healthy Aging Research Center, Taoyuan, Taiwan.

### 2.4. NMR Data Processing and Analysis

The raw ^1^H-NMR spectra were processed using NMRProcFlow online software to calibrate ppm, correct baseline, bucket spectra, and normalize data. Spectra bucketing was applied using intelligent bucketing and variable size bucketing. Chenomx NMR Suite 9.0 professional software (Chenomx Inc., Edmonton, AB, Canada) was then utilized to identify metabolites. According to previously established NMR data analysis methods [16], generalized log transformation (glog) and Pareto scaling were applied to the ^1^H-NMR spectra data. Partial least squares-discriminant analysis (PLS-DA) in MetaboAnalyst (version 5.0) was used for identifying the metabolites used for discrimination between the groups. Metabolites with a variable importance in projection (VIP) score ≥ 1.0 with a *p*-value < 0.05 were considered significant. Functional metabolic pathways were pictured based on the Kyoto Encyclopedia of Genes and Genomes (KEGG) database.

### 2.5. Statistical Analysis

Comparisons of baseline characteristics between groups with different cholesterol statuses were performed with univariable parametric and non-parametric tests as appropriate. Differences in metabolites were assessed by Mann–Whitney U test on the MetaboAnalyst web server. A false discovery rate (FDR) of 5% was applied for correcting for multiple comparisons. The correlation coefficients between blood metabolites and serum lipid profiles were calculated using Spearman’s rank correlation test in R software (Lucent Technologies, Murray Hill, NJ, USA, version 4.2.1). Random forest classifier models were used to rank features with the largest contribution toward metabolite and demographic data with the Boruta feature selection algorithm by a 20-fold stratified cross-validation testing procedure. Statistical analysis was performed using the Statistical Package for the Social Sciences (SPSS Statistics for Windows Version 20.0; Armonk, NY, USA). All statistical hypothesis tests were 2-tailed and a *p*-value < 0.05 was considered significant.

## 3. Results

### 3.1. Population Characteristics

A total of 125 subjects were enrolled into this study, including 42, 52, and 31 children with acceptable, borderline, and high levels of cholesterol, respectively. Table 1 shows the baseline characteristics among different cholesterol status groups. There was a significantly higher ratio of female children with high cholesterol levels. Lipid profiles including TG, LDL, HDL, Apo-A1, and Apo-B were significantly increased with increasing cholesterol levels (*p* < 0.01). However, no significant differences were observed in age, height, weight, and BMI in children with different cholesterol statuses.

### 3.2. Identification of Blood Metabolites for the Different Cholesterol Statuses

^1^H-NMR data from plasma samples were collected and analyzed. One thousand buckets varied across NMR spectrum, 93 buckets of which corresponded to 39 known metabolites that were identified using the Chenomx NMR Suite. PLS-DA was used to identify metabolites discriminated between groups, and the results are shown in Appendix A. Although the *p*-value of the permutation test between groups showed no significant differences, the PLS-DA score plots still characterized the differential expressing metabolites (Appendix A). Metabolites selected by using a cutoff of FDR-adjusted *p*-value < 0.05 are shown in Table 2 alongside the expression level of VIP score and fold change. Among them, eight metabolites, tyrosine, glutamic acid, ornithine, lysine, alanine, creatinine, oxoglutaric acid, and creatine, were significantly higher in both the high and borderline cholesterol level group than in the acceptable cholesterol level group (FDR-adjusted *p* < 0.05).

### 3.3. Correlation between Blood Metabolites and Lipid Profiles

Cholesterol level was strongly positively correlated with LDL-C and Apo-B levels, as well as HDL-C and Apo-A1 level separately (*p* < 0.01). By contrast, HDL-C level was negatively correlated with triglyceride level (*p* < 0.01). Tyrosine, oxoglutaric acid, ornithine, lysine, and glutamic acid showed strong positive correlations with cholesterol, as well as LDL-C and Apo-B level but not HDL-C and Apo-A1 (Figure 1A, *p* < 0.01).

Random forest regression models were performed to discriminate children with different cholesterol statuses based on the significantly differentially expressed metabolites only (Figure 1B) or a combination with baseline characteristics (Figure 1C). Among the five strongly cholesterol-correlated metabolites, glutamic acid and tyrosine appeared to have the highest importance in terms of different cholesterol status. However, potential confounding variables including age, sex, BMI, weight, and height were ranked in the lowest importance quartile.

### 3.4. Metabolic Pathway and Function Analysis

The metabolic functional pathways related to cholesterol level are shown in Table 3. Amino acid metabolisms, including arginine and proline metabolism, alanine, aspartate, and glutamate metabolism, and arginine biosynthesis, were significantly associated with cholesterol status (FDR adjusted *p* < 0.01). Notably, glutamic acid was involved in all these amino acid metabolic pathways. Figure 2 illustrates a composite representation of significant metabolites and their potential functional pathways, elucidating the proposed molecular mechanisms. Cholesterol-related metabolites associated with metabolic pathways appeared to link to TCA cycle metabolism.

## 4. Discussion

Hypercholesterolemia is commonly noted in adults with obesity or metabolic syndrome, and it is a serious health concern. Of note, there is increased prevalence of hypercholesterolemia in the pediatric population and several studies have reported the increased risk of further cardiovascular diseases in these children [17,18]. A cross-sectional survey of the prevalence of hypercholesterolemia revealed a significantly increased rate (from 6.2% to 13.8%) in school-age children in Taiwan from 1996 to 2006 [19], which is consistent with approximately 10~13% of children who suffered from hypercholesterolemia in two large national surveys in America about a decade ago [20,21]. However, in this study, up to 25% of 7-year-old children had high cholesterol levels (≥200 mg/dL). The changes in lifestyle and dietary habit in the last decade may particularly explain this finding; however, there remains an increase in public concerns regarding hypercholesterolemia in children. 

The total level of cholesterol varies by age and sex [22]. As in this study, a higher cholesterol level has been reported in girls including school-age children and adolescents [18,20,21]. During puberty, sexual and other physical maturation occurs as a result of hormonal changes; in parallel with this, there is a decrease in plasma total cholesterol levels in young adults [18]. After age 20, the total cholesterol level in plasma increases progressively and more rapidly in men, accounting for higher total cholesterol levels than their female counterparts [14]. Despite debate, sex hormones drive changes in regulating lipoprotein metabolism, playing a role in the hypercholesterolemia in children [23].

A lipid profile is a panel of blood tests used to assess several cholesterol health parameters. Total cholesterol is defined as the sum of HDL-C, LDL-C, and very low density lipoprotein (VLDL)-C and is strongly associated with these lipid parameters as in this study. Furthermore, LDL is a high-Apo-B-containing lipoprotein, whereas HDL is rich in Apo-A1; this is the reason why cholesterol was correlated with LDL-C and Apo-B and with HDL-C and Apo-A1 separately in this study. Apart from cholesterol, triglyceride is a type of fat converted by any excess calories from foods and mostly stored in fat cells for a longer-term energy source. A strong inverse correlation for triglyceride with HDL-C levels in this study further supports that hypertriglyceridemia leads to HDL-C consumption via increasing unstable TG-rich HDL particles, which are the more susceptible to clearance [24]. 

Several studies have targeted the metabolic mechanism of hyperlipidemia and amino acids have been reported as important endogenous signaling molecules of lipid metabolism [25,26]. In this study, glutamic-acid-related amino acid metabolisms were strongly associated with hypercholesterolemia. Glutamic acid, a by-product of branched-chain amino acid (BCAAs) catabolism, has been shown to be increased in obesity, especially in the visceral area [27,28,29]. Increased visceral fat accumulation is acknowledged as a risk factor for dyslipidemia through the development of insulin resistance [30], which contributes to the strong positive correlation between glutamic acid and cholesterol level in this study. 

Apart from glutamic acid, tyrosine, a nonessential amino acid synthesized in the body, was also an important amino acid strongly associated with hypercholesterolemia in this study. A longitudinal analysis revealed a relationship between tyrosine metabolism and insulin resistance in obese children [31], leading to the high level of cholesterol in plasma. In addition, many studies have clarified that dietary tyrosine causes hypercholesterolemia in rodents, probably through the ability of tyrosine to influence liver metabolism towards the increased synthesis of lipids [32,33,34].

Carbohydrate and amino acid metabolism are both confluent to the energy-producing TCA cycle, which is also a crucial step in the metabolism of fat [35]. Glucose is metabolized through glycolysis to form acetyl-CoA, which is a precursor of the biosynthesis to cholesterol. Clinically, a diet high in refined carbohydrates and added sugars can lead to weight gain with a risk for hypercholesterolemia [36]. Carbohydrate metabolism was observed to be involved in the metabolic pathway related to the different statuses of cholesterol level in this study, suggesting that the regulation of high carbohydrate intake may be responsible for the metabolism of lipids altering the cholesterol levels. 

The limitation of this study is primarily attributed to the small sample size and the relatively low sensitivity of NMR-based metabolomic analysis technique employed. However, ^1^H-NMR spectroscopy not only ensures a high analytical reproducibility but also a combination with the utilization of the PLS-DA method for metabolomic analysis has been shown to be efficient in identifying a wide range of metabolites associated with variations in cholesterol levels with a high degree of precision. Despite the lack of genetic confirmation for the potential existence of familial hypercholesterolemia, it is relatively low in prevalence and an extreme exclusion by family healthy history minimized its influence on the analytic results in this study. Furthermore, an age-matched design in this study eliminates the largest variability for metabolic profiles among different age groups. Most importantly, the selection of pre-pubertal age children in this study is expected to provide credible results for avoiding the influence of puberty on lipid profiles in relation to hypercholesterolemia.

In conclusion, this study found that as many as 25% of children had high levels of cholesterol (>200 mg/dL), highlighting the significance of hypercholesterolemia as a health concern among the pediatric population. Amino acid and carbohydrate metabolism are both significantly involved in lipid metabolism. However, glutamic acid and tyrosine are the most important amino acids associated with hypercholesterolemia. The metabolic pathways related to glutamic acid appear to be strongly associated with different cholesterol statuses, suggesting that amino acids play a crucial role in regulating lipid metabolism. Nonetheless, further functional studies in independent larger cohorts are warranted to investigate these associations more comprehensively.

## Figures and Tables

**Figure 1 nutrients-15-01726-f001:**
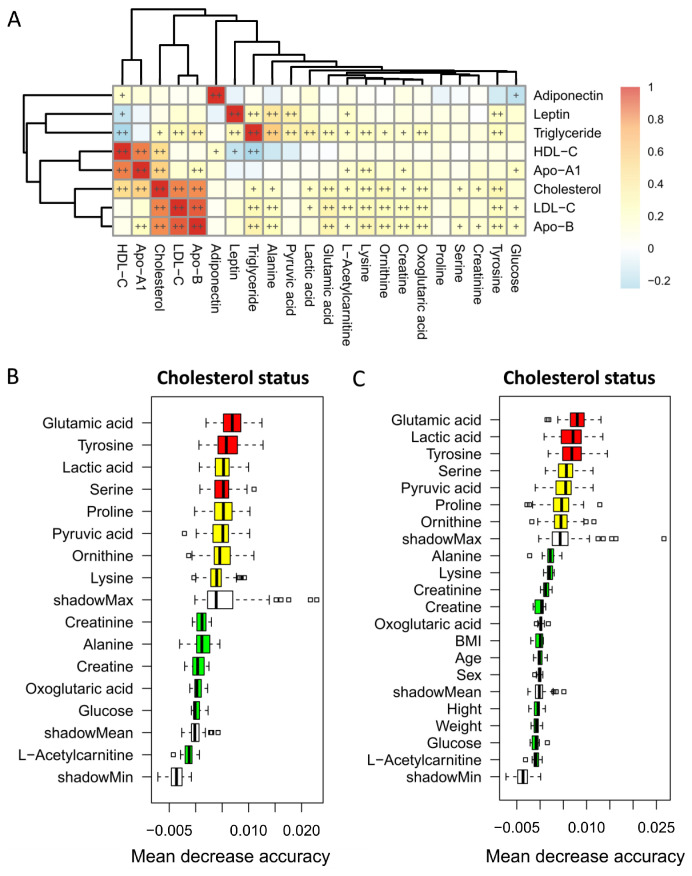
Heatmap of Spearman’s rank correlations of metabolites significantly differentially expressed within different statuses of cholesterol level with lipid profile levels (**A**). Color intensity represents the magnitude of correlation. + symbol means a *p*-value < 0.05; ++ symbol means a *p*-value < 0.01. Markers for differentiating children with different cholesterol statuses identified from random forest classifiers based on the significantly differentially expressed metabolites with metabolite profile only (**B**) or a combination with baseline characteristics (**C**). Markers are ranked in descending order of their importance to the accuracy of the model. The boxes represent the 25th–75th percentiles, and black lines indicate the mean.

**Figure 2 nutrients-15-01726-f002:**
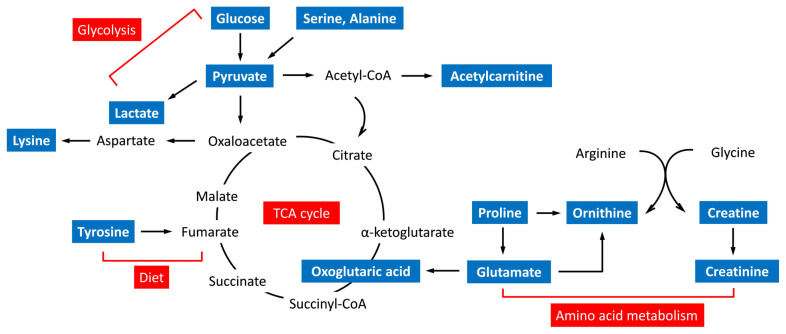
Schematic overview of the metabolic pathways associated with cholesterol-related metabolites.

**Table 1 nutrients-15-01726-t001:** Baseline characteristics of 125 children in relation to different status of cholesterol level.

Characteristics	Acceptable (*n* = 42)	Borderline (*n* = 52)	High (*n* = 31)	*p*-Values
Age (yr)	7.12 ± 0.25	7.20 ± 0.31	7.12 ± 0.06	0.683
Sex, male	27 (64.3%)	31 (60.8%)	11 (35.5%)	**0.031**
Height (m)	1.23 ± 0.06	1.24 ± 0.05	1.23 ± 0.06	0.829
Weight (kg)	24.92 ± 6.77	25.08 ± 5.31	23.96 ± 5.63	0.574
BMI (kg/m^2^)	16.33 ± 3.52	16.29 ± 2.50	15.82 ± 2.84	0.406
Obesity	5 (12%)	7 (13%)	6 (19%)	0.648
**Lipid profile**				
Cholesterol	151.64 ± 16.92	185.25 ± 8.48	223.72 ± 21.45	**<0.001**
TG	47.55 ± 22.68	57.60 ± 36.90	69.28 ± 34.86	**0.010**
LDL	82.92 ± 14.46	103.9 ± 14.92	130.28 ± 12.48	**<0.001**
HDL	52.33 ± 9.10	60.99 ±12.74	68.89 ± 13.22	**<0.001**
Apo-A1	134.64 ± 18.05	148.85 ± 18.42	165.48 ± 20.95	**<0.001**
Apo-B	56.95 ± 9.92	70.87 ±9.47	87.17 ± 11.02	**<0.001**
Adiponectin	21.62 + 12.02	20.53 ± 9.17	23.51 ± 10.30	0.275
Leptin	5.07 ± 7.03	6.25 ± 8.58	5.53 ± 5.93	0.571

Data shown are mean ± s.d. or number (%) of patients as appropriate. BMI, body mass index; m, meter; kg, kilogram; TG, triglyceride; LDL, low-density lipoprotein; HDL, high-density lipoprotein; Apo, apolipoprotein. All *p*-values < 0.05, which is in bold, are significant.

**Table 2 nutrients-15-01726-t002:** The VIP score and fold change of metabolites significantly differentially expressed between children with different cholesterol levels.

		High vs. Acceptable			Borderline vs. Acceptable			High vs. Borderline		
Metabolites	Chemical Shift, ppm (Multiplicity)	VIP Score *	Fold Change †	*p* ‡	VIP Score	Fold Change	*p*	VIP Score	Fold Change	*p*
Tyrosine	6.860−6.917(dt)	2.26	1.16	**0.000**	1.62	1.14	**0.004**	0.85	1.02	0.455
Glutamic acid	2.094−2.130(m)	1.47	1.08	**0.001**	1.37	1.10	**0.002**	0.70	0.98	0.774
Ornithine	3.041−3.075(t)	1.65	1.10	**0.004**	1.73	1.13	**0.001**	0.75	0.98	0.646
Lysine	3.010−3.030(t)	1.30	1.07	**0.007**	1.33	1.09	**0.003**	0.66	0.98	0.683
Alanine	1.455−1.490(q)	1.69	1.14	**0.013**	1.86	1.18	**0.002**	0.64	0.96	0.612
Creatinine	3.036−3.041(s)	1.16	1.06	**0.013**	1.26	1.08	**0.004**	0.60	0.98	0.571
Lactic acid	1.304−1.340(t)	1.79	1.14	**0.020**	0.99	1.12	0.168	1.10	1.02	0.434
Oxoglutaric acid	2.980−3.010(dt)	1.10	1.06	**0.024**	1.24	1.08	**0.005**	0.76	0.98	0.568
L-Acetylcarnitine	3.182−3.187(d)	1.22	1.08	**0.029**	0.87	1.06	0.079	0.99	1.01	0.648
Creatine	3.917−3.926(s)	1.18	1.08	**0.044**	1.31	1.11	**0.013**	0.68	0.97	0.661
Serine	3.937−3.957(q)	0.85	1.05	0.085	1.36	1.11	**0.005**	1.13	0.94	0.219
Glucose	4.622−4.656(d)	0.85	1.05	0.099	1.04	1.07	**0.026**	0.85	0.98	0.563
Pyruvic acid	2.358−2.372(s)	0.71	1.06	0.301	1.69	1.17	**0.005**	1.48	0.91	0.134
Proline	3.309−3.339(m)	0.60	1.04	0.359	1.89	1.18	**0.001**	2.00	0.88	**0.031**

* VIP scores were obtained from PLS-DA. † Fold changes were calculated by dividing the value of metabolites in children with high or borderline levels of cholesterol by those in children with acceptable levels and those in children with high levels of cholesterol by those in children with borderline levels. VIP, variable importance in projection; ppm, parts per million. Multiplicity, dt, doublet of triplet; m, multiplet; t, triplet; q, quartet; s, singlet; d, doublet. ‡ All FDR-adjusted *p*-values < 0.05; those in bold are significant.

**Table 3 nutrients-15-01726-t003:** Metabolic and functional pathway of metabolites associated with different statuses of cholesterol level.

Pathway Names	Total	Hits	Matched Metabolites	Raw*P*	FDR	Function
Arginine and proline metabolism	38	5	Creatine, Proline, Pyruvate Glutamic acid, Ornithine	1.700 × 10^−5^	0.001	Amino acid metabolism
Alanine, aspartate, and glutamate metabolism	28	4	Alanine, Glutamic acid, Pyruvate, Oxoglutaric acid	1.016 × 10^−4^	0.003	Amino acid metabolism
Arginine biosynthesis	14	3	Glutamic acid, Ornithine, Oxoglutaric acid	2.507 × 10^−4^	0.005	Amino acid metabolism
D-Glutamine and D-glutamate metabolism	6	2	Glutamic acid, Oxoglutaric acid	1.282 × 10^−3^	0.021	Amino acid metabolism
Glycolysis/gluconeogenesis	26	3	Pyruvate, Lactic acid, Glucose	1.669 × 10^−3^	0.023	Carbohydrate metabolism
Butanoate metabolism	15	2	Glutamic acid, Oxoglutaric acid	8.538 × 10^−3^	0.102	Carbohydrate metabolism
Citrate cycle (TCA cycle)	20	2	Oxoglutaric acid, Pyruvate	1.502 × 10^−2^	0.157	Carbohydrate metabolism

Total is the total number of compounds in the pathway; Hits is the number actually matched from the user uploaded data; Raw*P* is the original *p*-value calculated from the enrichment analysis; false discovery rate (FDR) is the portion of false positives above the user-specified score threshold. TCA, tricarboxylic acid.

## Data Availability

The data presented in this study are available on request from the corresponding author. The data are not publicly available due to their containing information that could compromise the privacy of research participants.

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
