# Peer review of "Metabolomics Analysis Reveals Molecular Signatures of Metabolic Complexity in Children with Hypercholesterolemia"

_nutrients, 2023, doi:10.3390/nu15071726_

Round 1

Reviewer 1 Report

Dear respected editor;

Regarding the article number nutrients-2275025, titled; Metabolomics Analysis Reveals Molecular Signatures of Metabolic Complexity in Children with Hypercholesterolemia

This study aimed to identify the metabolic signatures of different cholesterol status and their relationship with lipid profile in children using 1H-NMR spectroscopy. Metabolic pathways involved in the molecular mechanism of lipid metabolism will also be assessed, hopefully providing potentially health strategies for childhood hypercholesterolemia.

General points to be considered

-          Title: I think should be corrected to be in line with the findings of the study.  

-          Abstract: authors have to add the objective of the study clearly at the beginning of the abstracts.

The method part in the abstract can be improved and need to be rewritten completely with better way.

- This research is well-designed and written.

1.     Introduction:

Need to add more about justification and importance of the study.

2. Materials and Methods:

2.1 Study population Inclusion and exclusion criteria should be written and determined correctly.

3.      Results

Acceptable and it is well written.

4.      Discussion: well written

5.      Conclusion: could be improved.

References:

Well written according to the journal instructions.

Author Response

Submission ID: nutrients-2275025

Title: Metabolomics Analysis Reveals Molecular Signatures of Metabolic Complexity in Children with Hypercholesterolemia

Nutrients

Reviewer 1:

  1. Title: I think should be corrected to be in line with the findings of the study.  

Ans: As reviewer suggested, it is a good idea to have the title being in line with the findings. Since the molecular mechanism for hypercholesterolemia is quite complicated and a lot of significant metabolites and potential function pathways related to hypercholesterolemia were identified in this study. After considering it sincerely, the current title may be a nice way to represent the broad and complex findings using metabolomics approach for hypercholesterolemia in this study. We therefore humbly keep the current title and hope you find our rationale for the title choice reasonable.

  1. Abstract: Authors have to add the objective of the study clearly at the beginning of the abstracts.

Ans: As reviewer suggested, the objective (major aim) of the study has added in the abstract:

In the Abstract section,

 “Despite the importance of hypercholesterolemia in children, it is overlooked, and there are currently few metabolomics-based approaches available to understand its molecular mechanisms. Children originated from a birth cohort measured cholesterol levels were aimed to identify metabolites for the molecular biological pathways of childhood hypercholesterolemia. One hundred and twenty-five children……..”

  1. Abstract: The method part in the abstract can be improved and need to be rewritten completely with better way.

Ans: As reviewer suggested, more information has added in the method part in the abstract to achieve more completeness:
In the Abstract section,
“…Plasma metabolomic profiles were obtained from using 1H-nuclear magnetic resonance (NMR) spectroscopy and partial least squares-discriminant analysis (PLS-DA) was applied using MetaboAnalyst 5.0 platform. Metabolites significantly associated with different cholesterol status were identified, and random forest classifier models were used to rank the importance of these metabolite features related to cholesterol status. Their associations related to serum lipid profile and functional metabolic pathways associated with hypercholesterolemia were also assessed…..”

  1. Introduction:

Need to add more about justification and importance of the study.

Ans: The importance of this study has mentioned in the last paragraph of the Introduction section. To emphasize the justification and the importance of this study, some information has added in the Introduction and amended as below:

In the Introduction section,

“This study is aimed to identify the metabolic signatures of different cholesterol status and their relationship with lipid profile in children using 1H-NMR spectroscopy. A comprehensive investigation of using metabolomics-based approach and further functional pathway analysis can not only gain insight into the molecular mechanism of lipid metabolism, but also provide potential health strategies for childhood hypercholesterolemia.”

  1. Materials and Methods:

Study population Inclusion and exclusion criteria should be written and determined correctly.

Ans: As reviewer mentioned, the inclusion criteria had been written in the Materials and Methods already:

In section 2.1. Study Population,

“A prospective cross-sectional controlled study ………. Children recruited in a birth cohort launched in Taiwan and completed a 7-year follow-up period since birth were enrolled into this study………..”

As reviewer suggested, information regarding exclusion criteria for this study was added to the manuscript to clarify the study population:

In section 2.1. Study Population,

"Children with any chronic health conditions including congenital heart disease, congenital chromosome or genetic anomaly, and endocrine disorders such as diabetes or thyroid disease were excluded. In addition, to minimize the potential con-founding effects of genetic familial hypercholesterolemia, children with a family health history of early heart disease or hypercholesterolemia were excluded from the study.

Reviewer 2 Report

The present manuscript authored by Gu et al. reports that hypercholesterolemia is present among pre-pubertal population. They also found that lipid profile is associated to carbohydrate and amino acid metabolisms, being glutamic acid the most relevant. The study in its present form lacks completeness. I have the following comments:

-                 Statement of lines 45-47 is misleading. Other studies have already tackled the issue of hypercholesterolemia in children (e.g., Christensen et al., Atherosclerosis, 2017, among others).

-                 The novelty of this study is hampered by the fact that association of glutamic acid with cholesterol metabolism is widely demonstrated. Authors should specify how their work advances the scientific knowledge.

-                 Methods section is incomplete and several relevant details are missing. For instance, it is stated that children were recruited in a birth cohort and followed-up for a period of 7 years. When were blood samples obtained, at baseline or at 7 years of follow-up? Why authors did not perform a longitudinal comparison?

-                 Were the children included in this study screened for familial hypercholesterolemia by genetic testing?

-                 Sample size is too low. Further validation in independent larger cohorts is needed.

-                 How many biomarkers and/or metabolites were measured? Why fatty acids were not evaluated?

-                 Did authors include measures of genetic liability for metabolic alterations by using polygenic score-based modeling?

-                 One of the main findings of the study is that carbohydrate metabolism relates to cholesterol profile. It has been shown that childhood eating behaviors and lifestyle can impact blood metabolites in later stage of life such as adolescence. Have childhood physical activity level and dietary intake taken into account when analyzing this data?

-                 Authors should prove the detected metabolic pathways by functional studies.

Minor comments

-          Figure 2 should rather be used in Discussion / Conclusion than Results section.

Author Response

Submission ID: nutrients-2275025

Title: Metabolomics Analysis Reveals Molecular Signatures of Metabolic Complexity in Children with Hypercholesterolemia

Nutrients

Reviewer 2:

The present manuscript authored by Gu et al. reports that hypercholesterolemia is present among pre-pubertal population. They also found that lipid profile is associated to carbohydrate and amino acid metabolisms, being glutamic acid the most relevant. The study in its present form lacks completeness. I have the following comments:

  1. Statement of lines 45-47 is misleading. Other studies have already tackled the issue of hypercholesterolemia in children (e.g., Christensen et al., Atherosclerosis, 2017, among others).

Ans: As the reviewer noted, there are studies analyzed the metabolic features for childhood hypercholesterolemia, which were mainly focused on genetic familial hypercholesterolemia [1]. Our study was designed and aimed to analyze the metabolic profiles related to different cholesterol status in healthy children from a birth cohort. Children with a family health history of early heart disease or hypercholesterolemia were excluded from this study to reduce the interference of possible genetic familial hypercholesterolemia.

To describe our ideas more precisely, the statement is rewritten:

In section of Introduction,

“Dyslipidemia is known as one of the strongest risk factors for cardiovascular dis-ease with the increasing fact that atherosclerosis begins in childhood. In addition, there is growing attention over recognizing and treating of dyslipidemia in childhood to prevent or delay cardiovascular events in adulthood. Total cholesterol, the sum of non–high-density lipoprotein cholesterol (non–HDL-C) and HDL cholesterol (HDL-C), is the most often discussed factor for pediatric dyslipidemia. Clinically, a genetic pediatric hypercholesterolemia, the familial hypercholesterolemia, has been addressed [5]. However, the molecular mechanism of hypercholesterolemia in healthy children from a birth cohort remains few approached.”

  1. The novelty of this study is hampered by the fact that association of glutamic acid with cholesterol metabolism is widely demonstrated. Authors should specify how their work advances the scientific knowledge.

Ans: As kindly pointed out by the reviewer, previous studies have shown an association between glutamic acid and cholesterol metabolism. Our search revealed that most of these studies were conducted on animal models or cell model of adults [2-6]. Importantly, our study contributes to the understanding of the potential association between glutamic acid and cholesterol metabolism specifically in children. In addition to the finding above, as high as 25% of children had high levels of cholesterol (>200 mg/dL) in this study, highlighting the significance of hypercholesterolemia as a health concern among the pediatric population. Also, carbohydrate metabolism was found to be significantly involved in lipid metabolism.

Detail information regarding the novelty of this study has concluded in the Conclusion section of Discussion:

In the Discussion section,

“In conclusion, this study found that as high as 25% of children had high levels of cholesterol (>200 mg/dL), highlighting the significance of hypercholesterolemia as a health concern among the pediatric population. Amino acid and carbohydrate metabolism are both significantly involved in lipid metabolism. However, glutamic acid and tyrosine are the most important amino acids associated with hypercholesterolemia. Among them, metabolic pathways related to glutamic acid appear to be strongly associated with different cholesterol status, suggesting that amino acids play a crucial role in in regulating lipid metabolism. Nonetheless, further functional studies in independent larger cohorts are warranted to investigate these associations more comprehensively.”

  1. Methods section is incomplete and several relevant details are missing. For instance, it is stated that children were recruited in a birth cohort and followed-up for a period of 7 years. When were blood samples obtained, at baseline or at 7 years of follow-up? Why authors did not perform a longitudinal comparison?

Ans: The children from the cohort study were followed-up since birth. After a 7-year-follow-up, that is, when they were 7 years old, the blood samples were obtained. Since lipid levels vary during early adolescence, we conducted the test in their prepubertal period. A longitudinal comparison, exploring changes over time, is a valuable recommendation for our study. However, a longitudinal study is not initially designed due to inadequate follow-up period, and cannot be accessed and investigated at this moment. To describe precisely, the information regarding this has added and amended as follow:

In Method section,

2.1. Study Population

“A prospective cross-sectional controlled study was conducted to investigate the metabolomic profiles of blood in children with different status of cholesterol levels. Children recruited in a birth cohort launched in Taiwan and completed a 7-year follow-up period since birth were enrolled into this study. Blood samples were obtained at the age of 7 years old. Children with any chronic health conditions including………”

  1. Were the children included in this study screened for familial hypercholesterolemia by genetic testing?

Ans: The aim of our study is to investigate the molecular mechanism related to the different cholesterol levels in healthy children from a birth cohort. Familial hypercholesterolemia is an inherited genetic disorder that passed down through families and characterized by high cholesterol levels. The prevalence of familial hypercholesterolemia is about 1/200 to 1/250, and that in this cohort, only very few children had the family health history of early heart disease or hypercholesterolemia. Genetic testing for familial hypercholesterolemia was therefore not to be routinely screened under this circumstance. In addition, to reduce the influence of familial hypercholesterolemia on the analytic result, children with a possibility of the disease were excluded to the greatest extent possible from this study by a family health history.

Detailed information regarding this has added in the exclusion criteria:

In Method section,

2.1. Study Population,

"Children with any chronic health conditions including congenital heart disease, chronic lung disease, renal insufficiency, cerebral palsy, and endocrine disorders, diabetes or thyroid disease were excluded. In addition, to minimize the potential con-founding effects of familial hypercholesterolemia, children with a family history of hypercholesterolemia were excluded from the study.”

Also, limitation related to this information has added in the limitation for more information:

In Discussion section,

“The limitation of this study is primarily attributed to the small sample size and a relatively low sensitivity of NMR-based metabolomic analysis technique employed. However, 1H-NMR spectroscopy not only ensures a high analytical reproducibility, but also a combination with the utilization of PLS-DA method for metabolomic analysis has been shown to be efficient in identifying a wide range of metabolites associated with variations in cholesterol levels with a high degree of precision. Despite the lack of genetic confirmation for the potential existence of familial hypercholesterolemia, its relatively low prevalence and an extremely exclusion by family healthy history minimize its influence on analytic results in this study. Furthermore, an age-matched design in this study eliminates the largest variability of metabolic profiles among different age groups….”

  1. Sample size is too low. Further validation in independent larger cohorts is needed.

Ans: As the reviewer noted, we acknowledge the small sample size, and this information has mentioned in limitation already as follows:

In the limitation of Discussion section,

The limitation of this study is primarily attributed to the small sample size, and a relatively low sensitivity of NMR-based metabolomic analysis technique employed, and the lack of genetic confirmation of possible familial hypercholesterolemia for exclusion….”

As the reviewer suggested, the need for further validation in larger independent cohorts is a precious recommend, and it would serve as a commendable approach for our further planning for the studies in this issue. The idea has added to the Conclusion of our manuscript:

In Conclusion section,

“In conclusion, this study found that as high as 25% of children had high levels of cholesterol (>200 mg/dL), highlighting the significance of hypercholesterolemia as a health concern among the pediatric population. Amino acid and carbohydrate metabolism are both significantly involved in lipid metabolism. However, glutamic acid and tyrosine are the most important amino acids associated with hypercholesterolemia. Among them, metabolic pathways related to glutamic acid appear to be strongly associated with different cholesterol status, suggesting that amino acids play a crucial role in in regulating lipid metabolism. Nonetheless, further functional studies in independent larger cohorts are warranted to investigate these associations more comprehensively.”

  1. How many biomarkers and/or metabolites were measured? Why fatty acids were not evaluated?

Ans: A total of 39 metabolites were identified in this study by using 1H–NMR spectroscopy with TSP in D2O. Information regarding this has described in the Results already:

In the 3.2. Identification of Blood Metabolites for Different Cholesterol Status of Result section,

1H-NMR data of plasma samples were collected and analyzed. One thousand buckets varied across NMR spectrum, 93 buckets of which corresponded to 39 of known metabolites were identified using Chenomx NMR Suite……..”

By using 1H–NMR spectroscopy, lipid molecules often have very similar chemical structures, which can lead to signal overlap in NMR spectra. In addition, the major aim of this study was to identify the metabolic signatures of different cholesterol status. Precisely, lipid profiles including total cholesterol concentrations, low-density lipoprotein (LDL), high-density lipoprotein (HDL) levels, Apolipoprotein A1 (Apo-A1), and Apo-B levels were instead measured using an immunoturbidimetric assay with an automatic biochemical analyzer. A standard 1H–NMR spectroscopy with TSP in D2O TSP (3-trimethylsilylpropionic-2,2,3,3-d4 acid sodium salt) in D2O (deuterium oxide) was subsequently selected for a method for NMR analysis of organic compounds in this study.

Lipids are often insoluble in water, which is the solvent typically used in NMR experiments as in this study. Despite this, among the 39 metabolites identified in this study, few fatty acids including formic acid, acetic acid, isobutyric acid, and iso valeric acid were still can be identified. However, these fatty acids were not significantly differentially expressed between different cholesterol status and not shown in Table 2.

  1. Did authors include measures of genetic liability for metabolic alterations by using polygenic score-based modeling?

Ans: The aim of our study is to investigate the molecular mechanism related to the different cholesterol levels in healthy children from a birth cohort. The prevalence of genetic associated diseases is relatively low for a birth cohort study design, genetic testing for these children was not to be routinely screened. The analytic strategy of this study is therefore not initially designed using genetic-based method. The analysis of this study using polygenic score-based modeling cannot be performed at this moment.

  1. One of the main findings of the study is that carbohydrate metabolism relates to cholesterol profile. It has been shown that childhood eating behaviors and lifestyle can impact blood metabolites in later stage of life such as adolescence. Have childhood physical activity level and dietary intake taken into account when analyzing this data?

Ans: In this study, amino acid metabolism associated with childhood hypercholesterolemia appears to be a most robust correlation. Apart from this, the significant association between carbohydrate metabolism and cholesterol levels is also one of the notable findings in our study. Based on current medical knowledge, it is reasonable to infer that eating behaviors and lifestyle of these subjects may contribute to the carbohydrate metabolism. Unfortunately, information regarding physical activity level and dietary intake were not routinely recorded in this cohort study. To describe precisely, information regarding this has amended in the Discussion section as below:

In Discussion section,

 " Carbohydrate and amino acid metabolism are both confluent to the energy-producing TCA cycle, also a crucial step in the metabolism of fat. Glucose is metabolized through glycolysis to form acetyl-CoA, which is a precursor of biosynthesis to cholesterol. Clinically, a diet high in refined carbohydrates and added sugars can lead to weight gain risking for hypercholesterolemia. Carbohydrate metabolism was observed to be involved in the metabolic pathway related to different status of cholesterol level in this study, suggesting that the regulation of high carbohydrate intake may be responsible for the metabolism of lipids altering the cholesterol levels.”

  1. Authors should prove the detected metabolic pathways by functional studies.

Ans: As the review suggested, the functional studies to prove the detected metabolic pathways should base on clear-noted metabolites. More independent larger cohort should be analyzed for a valid finding of such metabolites. Therefore, in the current stage, functional studies were not yet performed. We greatly appreciate the reviewer's constructive feedback, and it will undoubtedly be considered as we continue to pursue future investigations in this field. As mentioned in the manuscript:

In the Conclusion section,

“In conclusion, this study found that as high as 25% of children had high levels of cholesterol (>200 mg/dL), highlighting the significance of hypercholesterolemia as a health concern among the pediatric population. Amino acid and carbohydrate metabolism are both significantly involved in lipid metabolism. However, glutamic acid and tyrosine are the most important amino acids associated with hypercholesterolemia. Among them, metabolic pathways related to glutamic acid appear to be strongly associated with different cholesterol status, suggesting that amino acids play a crucial role in in regulating lipid metabolism. Nonetheless, further functional studies in independent larger cohorts are warranted to investigate these associations more comprehensively.”

Minor comments

  1. Figure 2 should rather be used in Discussion / Conclusion than Results section.

Ans: In this study, we have identified several cholesterol-associated metabolites and their potential metabolic pathways of molecular mechanisms for childhood hypercholesterolemia. It will be very nice to have a summary and an overview of our findings. Figure 2 mainly is to elucidate the proposed molecular mechanisms for childhood hypercholesterolemia by using the findings originated from the results of our analyses. Based on this, we therefore humbly decide to keep the Figure 2 in the Results section and hope you find our rationale for this choice reasonable. To describe precisely, information regarding Figure 2 has amended and shown below:

In the Results section,

3.4. Metabolic Pathway and Function Analysis

“The metabolic functional pathways related to cholesterol level are shown in Table 3. Amino acid metabolisms, including arginine and proline metabolism, alanine, aspartate and glutamate metabolism, and arginine biosynthesis, were significantly associated with different cholesterol status (FDR adjusted P < 0.01). Notably, glutamatic acid was involved in all these amino acid metabolic pathways. Figure 2 illustrates a composite representation of significant metabolites and their potential functional pathways, elucidating the proposed molecular mechanisms. Cholesterol-related metabolites associated metabolic pathways appeared to link to TCA cycle metabolism.”

Thanks for your precious comments and suggestions. Hope these responses could answer all your questions. Thanks again!!!

Reference used in this response to reviewers

  1. Christensen, J.J.; Ulven, S.M.; Retterstøl, K.; Narverud, I.; Bogsrud, M.P.; Henriksen, T.; Bollerslev, J.; Halvorsen, B.; Aukrust, P.; Holven, K.B. Comprehensive lipid and metabolite profiling of children with and without familial hypercholesterolemia: A cross-sectional study. Atherosclerosis. 2017, 266, 48-57.
  2. Adil, M.; Kandhare, A. D.; Ghosh, P.; Venkata, S.; Raygude, K. S.; & Bodhankar, S. L.. Ameliorative effect of naringin in acetaminophen-induced hepatic and renal toxicity in laboratory rats: Role of FXR and KIM-1. Renal Failure. 2016, 38, 1007–1020.
  3. Hassan Abd; E. R., Al-Yamani; M. A. S., & Sayrafi M. A.. Effect of Saudi Propolis on hepatitis male rats. Journal of Nutrition and Food Science. 2017, 7, 2.
  4. Wan Saidatul Syida, W. K.; Noriham, A.; Normah, I.; & Mohd Yusuf, M. Changes in chemical composition and amino acid content of soy protein isolate (SPI) from tempeh. International Food Research Journal. 2018, 25(4), 1–6.
  5. Liu, Y.; Yang, J.; Lei, L.; Wang, L.; Wang, X.; Ma, K. Y.; Yang, X.; & Chen, Z.-Y. Isoflavones enhance the plasma cholesterol-lowering activity of 7S protein in hypercholesterolemic hamsters. Food & Function. 2019, 10(11), 7378–7386.
  6. Mohd Rosmi, N.S.A.; Shafie, N.H.; Azlan, A.; Abdullah, M.A. Functional food mixtures: Inhibition of lipid peroxidation, HMGCoA reductase, and ACAT2 in hypercholesterolemia-induced rats. Food Sci Nutr. 2021, 9, 875-887.

Round 2

Reviewer 2 Report

The present manuscript authored by Gu et al. has partially addressed the comments. Neither validation / functional studies were performed nor data on dietary and physical activity habits added. Only few changes in the text were made. While the manuscript is not totally complete from my point of view, its potential interest on prevention of hypercholesterolemia in children favors the recommendation for acceptance.